# Emotions in Hospital School Professionals: An Approach from Cognitive Linguistics

**DOI:** 10.3390/children9091395

**Published:** 2022-09-15

**Authors:** Álvaro Javier González Concha, Ana Andrea Gajardo Rodríguez, Macarena del Pilar Gallardo Gómez

**Affiliations:** 1Department of Arts and Letters, Faculty of Education and Humanities, Universidad del Bío-Bío, Brasil 1180, Chillán, Chile; 2Department of Educational Sciences, Faculty of Education and Humanities, Universidad del Bío-Bío, Brasil 1180, Chillán, Chile; 3Department of Business Management, Faculty of Business Sciences, Universidad del Bío-Bío, Avda. Avenue Andrés Bello 720, Chillán, Chile

**Keywords:** emotions, hospital schools, cognitive linguistics, metaphors, hospital pedagogy

## Abstract

The question that frames this work is: What evidence of the emotional impact of working in hospital schools is found in the metaphorical language used by the professionals who work there? The purpose of this research is to contribute to the understanding of the emotions of professionals working in these schools. The methodology used is based on a qualitative approach and is framed within the descriptive paradigm, approached from linguistics and cognitive metaphor The metaphorical expressions analyzed were extracted from the presentations and dialogue tables (of professionals from 12 countries) held during the I International Seminar on Hospital Pedagogy organized by the Universidad del Bío-Bío (Chile) in May 2021. A total of 24 metaphors were analyzed and 13 are presented in this article. The metaphors found allowed us to identify conceptual domains of origin such as: “war”, “trip”, “heat”, “field” (among others), which allowed us to understand the emotional effects that work in hospital schools has on professionals. The results obtained allow us to provide inputs to focus in a more pertinent way on the preparation of these professionals in the control of their emotions for multidisciplinary work relationships and with children and their families.

## 1. Introduction

When we speak of metaphor, one of the first associations that the brain makes is with literature; suggesting that metaphor is an analogical inference and, as such, starts from the similarity of two or more things in one or more aspects to conclude the similarity of those things in some other aspect [1].

One of the most comfortable currents is constructivism. According to this position, reality is not independent of the observer [2]. That is, each person possesses their own reality and, therefore, there are no realities that are more real than others. Humanists are also comfortable with metaphors. They are probably so comfortable with the use of metaphors because they rely much more on literature than any other psychological current.

Working with metaphors fosters imagination and creativity. As Azzollini and Gonzalez [3] point out, during the problem-solving process, an analogical–metaphorical understanding can either constitute a solution, initiate the path to a solution, or substantially change the approach to the problem. In short, metaphors can constitute heuristic springboards.

Likewise, metaphors allow contact with the expression of emotions [4] and these will depend on the evaluation you make of this announcement as it may represent, for example, a threat, a challenge, or a loss, among others. While some people tend to collapse, others express what is called self-affirmation and face an illness as a challenge, and there are even those who may deny the existence of any pathology.

The diagnosis itself represents an alteration in the biological functions of the person, in addition to engendering emotional and social impacts such as isolation, loneliness, interruption of a professional or personal project in the sufferer and their family members, and especially in their main caregivers [5].

It is necessary to stop for a moment and ask ourselves: What are emotions for? Does feeling emotions help to cope with the disease? Do they hinder us in this coping? However, the main function of emotions is to facilitate adaptation to the environment and the survival of the organism [6].

Regarding cognitive linguistics, which is the theoretical basis for this work, we can safely say that it began as a theoretical approach in the 1980s and that George Lakoff and Ronald Langacker, both American generativist linguists, are considered its precursors. As a scientific paradigm, cognitive science is heterogeneous, as it combines different fields such as artificial intelligence, neuroscience, psychology and linguistics, among others, as a way to delve into the complex world of human cognition and intelligence associated, for example, with the cognitive processing of language [7], which is the focus of this paper.

Our approach to cognitive linguistics is through metaphor, which we understand as a “figurative” use of language. Metaphor traditionally, and until before Lakoff and Johnson were considered as part of literary language and associated with the “embellishment” of language, that is, as part of the world of the rhetorical and associated with a rather superficial level of language and not as part of the structure of thought.

“We have come to the conclusion that metaphor, on the contrary, permeates everyday life, not only language, but also thought and action. Our ordinary conceptual system, in terms of which we think and act, is fundamentally metaphorical in nature” [1] (p. 35).

Cognitive linguistics breaks with two paradigms; the first is that metaphor is recognized as part of everyday language and of generalized use in human language, and not only as part of literary language (add Lakoff and Johnson); and the second is that metaphor is no longer understood as part of a superficial level (of the lexicon), but rather as a relationship between conceptual domains, in the conceptual system, which, through a process of correspondences, generates the metaphorical expressions that we use every day to communicate. As Croft and Cruse point out, metaphor is rather the result of a special process of reaching or conceptualizing meaning [8]. Cognitive linguistics understands metaphor as a process or mechanism that allows us to understand and express ourselves about situations that are complex, using more everyday and familiar concepts. For example, to refer to a love relationship we use expressions such as “we embarked on this relationship together”, “unfortunately, we are coming to the end of our journey”, “our relationship has been on a bumpy road”. In this case, the metaphor is DEATH IS A JOURNEY.

Following Lakoff and Johnson [1] the metaphor, in its internal structure, is analyzed in the following way: we will call source domain the domain from which the concepts we use in the metaphorical expression come from (the domain that lends its concepts): in the example of the metaphor DEATH IS A JOURNEY the source domain is JOURNEY since to this conceptual domain belong the concepts: “embark” or “potholes”. The target domain (some authors call it “destination domain”), on the other hand, is the domain on which the concepts coming from the source domain are superimposed, in the case of our example, the target domain is LOVE. Following Cuenca and Hilferty:

“Metaphor is thus understood as the projection of some concepts from one conceptual domain (the source domain) to another conceptual domain (the target domain).” [8] (p. 25)

By identifying the conceptual domains (origin and target), we can identify the relationships that are established between both domains, in a process that Lakoff and Johnson call “mapping” and that allows us to establish the web of relationships that is established, where this web arises from metaphorical expressions. In the case of the metaphor LOVE IS A JOURNEY, let us consider the following metaphorical expressions:

“our relationship has gone through many bumps”.

“we have reached the end of our journey”

“it caused me great sadness to learn that you took a detour in our relationship”.

The mapping, in this case, is carried out as follows:

The Table 1 that shows the mapping is the one that allows us to identify the conceptual relationships established in the conceptual system of people, when using these expressions. In this manner, we can understand the way we conceptualize love and thus the mental associations we construct.

We can find, to explain in more detail, other metaphors, such as: DISCUSSIONS ARE WARS, a widely analyzed metaphor, in which we can deduce that, when it comes to discussions, people tend to visualize them as a war since, for example, expressions such as “you attacked me with a battery of arguments”, “your defense in the discussion was weak”, “I will use heavy artillery in the debate”, “with the last argument you threw a missile” are used. In these expressions, we can see that terms such as “you attacked”, “defense”, “heavy artillery” and “missile”, evidently belong to the conceptual domain WAR; however, it is very common that we find them in metaphorical expressions associated with DISCUSSIONS. Normally, the conceptual domain LOVE (as a target domain), is a domain that is related to several target domains, as we saw in a previous example related to the source domain “travel”; however, it is also related to other source domains, generating other metaphors: LOVE IS FIRE, with expressions such as “where fire there was ashes remain”, “your love is burning”; LOVE IS A THEFT, in which metaphorical expressions appear such as “you stole my heart”, “thief of love”; in the case of the metaphor LOVE IS A JAIL, we find expressions such as “I am a prisoner of your love”, “the chains of your love hold me by your side”. In these examples, we can see how the metaphorical expressions we use in everyday language are generated from the relationship of two conceptual domains in the conceptual system, which allows us to understand metaphorical expressions that perhaps we had never heard before”. And we have found a way to begin to identify in detail what exactly are the metaphors that structure the way we perceive, think and act”.

Another relevant aspect is constituted by the existing relationship between language and culture, if we understand that language is part and vehicle of culture, then we assume that it reflects the worldview of the human group that speaks a given language. That is to say, those who speak Spanish as their mother tongue, for example, not only speak Spanish, but also think and structure the world from the perspective of Hispanic culture. Of course, the metaphor does not escape this relationship:

“The most fundamental values in a culture will be consistent with the metaphorical structure of the fundamental concepts in it. For example, consider some cultural values in our society that are consistent with our specializing metaphors: UP-DOWN and whose opposites would not be consistent.” [1] (p. 55).

In effect, the orientation metaphor UP-DOWN reflects a cultural pattern associated with the fact that the good, the better, that which is “more” is related to UP and the worse, the bad, that which is considered “less”, is associated with DOWN. Hence, an expression such as “Francisco is above you” (above) is understood as meaning that he is “better”, in no way would an expression like that be understood in the sense that Francisco is “less” or “worse”.

If we understand and accept that metaphor is associated with culture, then the passage from metaphor to action becomes somewhat obvious, the way in which our thinking is structured defines the way in which we act:

“Metaphors can create realities, especially social realities. A metaphor can thus become a guide for future action. These actions will, of course, conform to the metaphor. This will in turn reinforce the metaphor’s ability to make experience coherent. In this sense, metaphors can be fulfilling prophecies.” [1] (p. 188).

The choice of the cognitive metaphor model as an approach to the study of emotions in professionals working in the field of hospital pedagogy seems valid to us, since a metaphor “can become a guide for actions”, i.e., by studying the metaphors used, the behavior of professionals can be better understood. Indeed, the relationship between language, thought, and action has been sufficiently evidenced by cognitive linguistics studies, therefore, the choice of the metaphorical analysis model used in this study is fully justified owing to the large existing literature.

In the field of hospital pedagogy, the situation of emotional stress to which teachers and, of course, health personnel are subjected, generate actions and decisions relevant to the recovery of the treated child or adolescent. The effect that the language used has on the patients’ families and on the children themselves is evident. If we study the metaphors and analyze the relationships established in the conceptual system (between the target domain and the source domain), then we can better understand the way in which the people involved process, in cognitive and also affective terms, the situations experienced on a daily basis. Another important aspect is that if the people who accompany hospitalized children (health personnel, hospital school teachers and families) are aware of the effects that the metaphorical language used has on the child, then the use of metaphorical expressions that favor the understanding of the disease, its treatment and the recovery process in general could be favored.

## 2. Materials and Methods

### 2.1. Objectives and Research Design

The general objective of this research is to contribute to the understanding of the emotions of professionals working in hospital schools and the specific objectives proposed for this research are: (1) To identify the metaphorical expressions present in the speakers, exhibitors and participants of the dialogue tables, which took place during the I International Seminar on Hospital Pedagogy. (2) To analyze the metaphorical expressions of the speakers, exhibitors and participants of the dialogue tables, in the context of the International Seminar on Hospital Pedagogy. (3) To demonstrate the relationship between the metaphors used and the emotions involved in hospital school professionals.

This research uses the qualitative approach that Herrera [9] defines as “a category of research designs that extract descriptions from observations that take the form of interviews, narratives, field notes, recordings, audio transcriptions, written records of all kinds, photographs or films, and artifacts”; and is framed within the descriptive paradigm, which Cazau defines as “In a descriptive study, a series of questions, concepts or variables are selected and each is measured independently of the others, with the aim, precisely, of describing them. These studies seek to specify the important properties of persons, groups, communities or any other phenomenon” [10] (p. 27).

### 2.2. Participants

The interventions of 36 speakers who participated in the I International Seminar on Hospital Pedagogy, organized in Chile by the Universidad del Bío-Bío and the Ministry of Education of Chile, with the participation of 16 Latin American countries, were analyzed. The presentations were analyzed and the metaphorical expressions that were found were collected, of which, in a second selection process, only those relevant to the field of hospital pedagogy were left out. The speeches and presentations analyzed were those of the thirty-six speakers, exhibitors and participants of the seminar’s dialogue tables. These speakers, exhibitors and participants in the dialogue tables came from the following countries: Argentina, Bolivia, Ecuador, Uruguay, Costa Rica, Peru, Chile, Brazil, Venezuela, Paraguay, Spain, Mexico, Guatemala, Colombia and Panama.

The data collection procedure was performed considering all metaphorical expressions associated with hospital pedagogy that appeared in the interventions mentioned in the previous paragraph. No metaphorical expression that met this criterion was left out of the study, in order to cover all the underlying metaphors. This avoided any bias in the selection procedure and ensured that all expressions associated with hospital pedagogy were considered.

### 2.3. Instruments

Each of the presentations and round tables developed in the seminar were recorded in videos that remained from the transmission that was carried out through the Zoom platform, Youtube and Facebook, and the link was made through the Reuna Chile platform.

For the analysis of metaphorical expressions, the Lakoff and Johnson model was used, based on the mapping of conceptual domains involved in the metaphor (origin and goal).

### 2.4. Ethical Considerations

When registering for the seminar, each of the speakers, exhibitors and participants in the round tables were asked to sign a consent form for the use of images and information.

### 2.5. Data Analysis

A metaphorical analysis will be made from the perspective of cognitive linguistics of Lakoff and Johnson [1] and of the language used by the experts in hospital pedagogy at the I International Seminar on Hospital Pedagogy in Latin America: realities and challenges of the future, for which the lectures, presentations and dialogue tables developed were compiled. The analysis consisted of four stages, which are mentioned below:

Stage One: The conferences, expositions and dialogue tables carried out in the seminar were analyzed and metaphorical expressions were identified. As a result of this analysis, 70 metaphorical expressions were found, although not all of them were directly related to hospital pedagogy.

Stage two: Of the 70 metaphorical expressions found, those related to hospital pedagogy were selected, leaving 46 expressions.

Stage three: The 46 selected expressions were analyzed with the model proposed by Lakoff and Johnson [1]. From this analysis, 24 metaphors emerged, from which the target and origin domains were identified, and the ontological correspondence mapping was also carried out.

Stage four: The results were discussed to establish the relationship between the metaphors found and how they are projected to the vision that hospital pedagogy professionals carry out on a daily basis.

## 3. Results

### Metaphorical Analysis


Metaphor: THE HOSPITAL CLASSROOM IS A JOURNEY


According to the RAE, we can take two meanings for the word, journey. Firstly, it is a transfer that is made from one place to another by air, sea or land. In this case, both the work team and the children who are part of the hospital classroom work to achieve their objectives. This represents a process similar to a journey in which one has to move forward to reach a place (achieve the objective) (Table 2).


Metaphor: TO WALK IS TO LEARN


The meaning of learning, according to the dictionary of the Spanish language, is to acquire knowledge of something through study or experience. Therefore, it is not related to the action of walking, and it is for this reason that WALKING IS LEARNING is considered a metaphor. However, in this case it is used to communicate that students are continuing with their studies and with their lives (Table 3).


Metaphor: LIFE IS WAR


The word war, according to the Royal Spanish Academy, is defined as: armed struggle between two or more nations or between sides of the same nation. In this case, there is no concordance with the word life, which is why the expression LA VIDA ES GUERRA (LIFE IS WAR) is considered a metaphor. However, in this case, the metaphorical expressions relate to the way in which the people who belong to the hospital classroom see their lives, as if they were in a real war (Table 4).


Metaphor: EMOTIONS ARE CONTAGIOUS DISEASES


To understand the meaning of the expression contagious diseases, we must look for its definition separately, where disease is a more or less serious alteration of health, and contagious, according to the Dictionary of the Spanish language in its 23rd edition, is “said of a disease: that sticks and communicates by contagion. That sticks or spreads easily”. This definition neither as a whole nor separately agrees with the meaning given to the word emotion. However, in this case emotions can spread in the same manner as a contagious disease (Table 5).


Metaphor: TO COLLABORATE IS TO HOLD HANDS


To forge the idea of holding hands, we must first understand that the hand is a part of the human body attached to the extremity of the forearm and includes from the wrist to the tips of the fingers, and to hold is defined as to grasp or grasp with the hand. Despite the fact that the word collaborate and the expression hold hands are not similar in meaning, in this case they are related through a metaphor since the intention is to communicate that when one takes another person by the hand it is to help or collaborate (Table 6).


Metaphor: LABOR IS AN ARTICULATED MACHINE


The idea of an articulated machine can be conceived by separating the two words that compose it. On the one hand, machine refers to a set of devices combined to receive a certain form of energy and transform it into a more suitable one, or to produce a certain effect, and on the other hand, articulated denotes something that has articulations, there is a joint between two rigid parts that allows relative movement between them (Table 7).


Metaphor: THE HOSPITAL CLASSROOM IS A FIELD


A field is defined as a large piece of unpopulated land or farmland, so its meaning does not apply to what we know as a hospital classroom in any sense. However, we can perceive it as a field in which there are many objectives to accomplish (Table 8).


Metaphor: AFFECTION IS WARMTH


Warmth is commonly known as the sensation experienced at high temperatures and culturally, warmth is associated with positive affection that can be given to people (Table 9).


Metaphor: TO IMPROVE IS TO ILLUMINATE (CENTER)


To illuminate is to illuminate, to give light. In the case of this metaphor, it projects the idea that the work done in the hospital classroom generates a positive effect on the care of hospitalized children and is highlighted for that reason (Table 10).


Metaphor: LIFE IS A PLAY


A play can be defined as any intellectual product in science, literature or the arts, especially one of some importance, and a theater as a place where an action is performed before spectators or participants, or the art of composing dramatic works, or of representing them. Seeing life as a play provides a broader view of the world and, above all, allows one to analyze the consequences of what people do, as if they were on a stage (Table 11).


Metaphor: THE SYSTEM IS A PERSON


The meaning of person is: individual of the human species/man or woman whose name is ignored or omitted. System is understood as: set of rules or principles on a subject rationally linked together (Table 12).


Metaphor: TEACHERS ARE TOOLS


According to the RAE, tools are an instrument, usually made of iron or steel, with which artisans work. By saying that teachers are tools, they are seen as an object that generates positive changes in students or “modifies” them (Table 13).


Metaphor: HEALTH IS AN OBJECT


An object is a thing, an inanimate object, as opposed to a living being. Health is more of a state, however, by conceiving of it as an object, we then become more aware that that state can change from good to bad if that object is “lost” (Table 14).

## 4. Discussion

The strength that cognitive linguistics has achieved is evident; cognitive science has been enriched with this contribution and today its scope is very generous. In this field, the incorporation of metaphor, from the perspective of cognitive linguistics, in behavioral psychology is increasing [11], as well as in psychoanalytic thought: “Freud claimed that thought in images was closer to the unconscious than thought in words” [12] (p. 35). González de Requena points out that “Regarding the mental, in general, it has been observed how the various metaphorical projections have contributed to forge different modes of self-consciousness and self-understanding of human activity.” [13] (p. 47). Seen in this way, then, the metaphors we use in our daily communication project a part of the self-understanding of what we do and how we relate to the world. In the field of hospital pedagogy, through the analysis of the metaphors found, we will be able to possess information on how the professionals of hospital schools who participated in the First International Seminar on Hospital Pedagogy, held in 2021 in Chile, understand their role in this field, and the construction of the world they carry out.

In the metaphors found, the vision highlights that work in hospital schools is fundamentally collaborative and teamwork is evident. The metaphors that reflect this are the following:

1. THE HOSPITAL CLASSROOM IS A JOURNEY

2. WALKING IS LEARNING

5. TO COLLABORATE IS TO HOLD HANDS

6. WORK IS AN ARTICULATED MACHINE

In metaphor 1, if we analyze the mapping, we will realize that the idea of the “journey” implies a journey of several actors; the hospitalized child does not travel alone, this is made explicit in the metaphorical expression: “That all the services agree, that all the staff agree and then we walk towards the same point”. The “journey” directly alludes to the fact that the work being done has stages and is a process in which we work collaboratively. This same idea is reflected in metaphor 2, which shows a concern for the children’s learning, also understood as “moving forward” on a journey.

In metaphors 5 and 6, what is mainly reflected is the idea of collaborative work, the image of “holding hands” (in metaphor 5) implies help and collaboration to “walk” or “move forward” in this journey. The following metaphorical expression shows this: “Teachers work hand in hand with parents or caregivers with the pediatrician, and head nurses”, this expression directly alludes to the importance of collaborative and multidisciplinary work among the actors involved in the physical and emotional recovery of children. In metaphor 6, the idea of an articulated machine alludes to the same representation as metaphor 5, namely of collaborative work. However, it pays special attention to the fact that it must be a very coordinated work, just as the pieces of a machine are coordinated: “We have a very articulated work with the health secretariat now in these times of pandemic”.

Undoubtedly, the management of emotions in the field of hospital pedagogy is a relevant topic, and it should therefore be of no surprise to us that there are metaphors that allude to this aspect. Indeed, metaphors 4 and 8 are evidence of this:

4. EMOTIONS ARE CONTAGIOUS DISEASES

8. AFFECTION IS WARMTH

In metaphor 4, the metaphorical expressions reveal an awareness that the emotions shown have an effect on the other people in the hospital school. The fact that emotions are “contagious” gives us this idea: “The hospital classroom also spreads that joy of continuing to work with the children”. In the case of metaphor 8, the idea that emotions are temperature carries over to this metaphor, if we analyze the expression: “On the other hand, it touches that emotional fiber that we must have, which gives warmth to health personnel” we will notice that “giving warmth to health personnel” implies the transfer of a positive feeling or emotion, the positive in terms of emotions relates to warmth and the negative to cold. It is for this reason that in this metaphor we speak of “warmth”, that is, a feeling of “positive affection” as opposed to “coldness”, which is associated to a feeling of apathy or also to the absence of feelings of affection.

Regarding metaphor 3: LIFE IS A WAR, it undoubtedly projects a way of viewing the hospital school as a place of tranquility and security, in the midst of a space in which death, pain and suffering is a constant. In expressions such as “The little school is a refuge”, it is evident that the people who work there perceive it as a place that provides security and tranquility, a welcoming space. Another relevant aspect that emerges from this metaphor can be seen in the expression “From the trench that each one approaches in the hospital”, in which the idea is made explicit that there is a permanent “war” and that the areas or specialties in which each professional works are perceived as the “trenches” from which they work to support the recovery of the health of hospitalized children. Undoubtedly, the daily coexistence with sick children and adolescents, the pain that these children often suffer and the emotional and even financial effects that these illnesses cause, are perceived as the “war” that the professionals who work there must endure on a daily basis.

The metaphor 10 LIFE IS A THEATER PLAY, alludes to the fact that in the process of treating the disease, the child or adolescent is the center of everything and that the decisions and actions of the professionals involved are centered on the child or adolescent. We can observe this in the expression “The health-disease process also has a special characteristic where he is the actor, the protagonist” in which the patient/student is seen as the “protagonist actor”, therefore, there is a clear awareness that both the treatment and the educational process are centered on the student. This takes on special relevance in the context of hospital pedagogy, since it is a multidisciplinary team that intervenes and that, effectively, must design its strategies and decision making according to the patient/student.

Another metaphor worthy of mentioning is HEALTH IS AN OBJECT, since by conceiving health, which is a state, as an object, one becomes more aware that this state can either improve “recovering the object” or deteriorate “losing the object”, as can be seen in the expression “Many of them experienced physical loss of their family members and also obviously loss of health”. When health is perceived as an object, it is easier to understand the concept of “taking care of it” and, therefore, to make the appropriate decisions to do so, just as it is easier for a child to conceive of health as an object than as a “state”.

In the six steps proposed for working with metaphors in therapy [14], the third says: “Expand the metaphor: At this point the client is invited to give the associations produced by the metaphor (the emotions and images it arouses)”. This approach provides challenges and possibilities that extend even further than those addressed in this article, and that concern the establishment of a direct dialogue with the people who use the metaphors, in order to search for and discover with greater precision the relationships that are established.

The contribution of this work to hospital pedagogy lies in the management of emotions, since it allows a more objective approach, through the study of language, to the study of the impact that the work carried out in this field has on professionals. This facilitates an understanding of the emotions involved and manifested through metaphorical language, considering the relationship between language–thought and action. Studies such as the one presented in this article, open up a space for work that has seldom been explored in hospital pedagogy (cognitive linguistics) and can become an interesting contribution to the study of the management of emotions, especially if a study such as this one is conducted with the families of hospitalized children and young people. A projection of this type of work would facilitate the understanding of the way in which families express their emotions and, in this way, discursive strategies could be generated on the part of professionals to facilitate communication with them. It seems undeniable that a projection of future work that addresses, from the cognitive metaphor, the approach to the representation of the reality of hospital pedagogy, projected through the metaphors of those involved in it, would lead not only to a better understanding of the cognitive and psychoemotional processing of this reality, but also to a strengthening of the preparation process that professionals in this area require.

In addition to the above, a similar work, developed with the families of the patients/students, bears interesting projections, as psychoemotional support, at the moment of facing the hospitalization of a son, brother, nephew, etc.

## Figures and Tables

**Table 1 children-09-01395-t001:** Love is a Journey.

Origin Domain Love	Target Domain Journey
End of trip	Termination of the relationship
Potholes	Difficulties
Bypass	Infidelity

**Table 2 children-09-01395-t002:** The Hospital Classroom is a Journey.

The hospital classroom also spreads the joy of continuing to work with the children and continuing to move forward with all our projects.	To move forward	To carry out the projects given within the hospital classroom.
We are working very closely with the Ministry of Health and we are trying, we have already taken several steps between the Ministry of Health and the Ministry of Education.	Advance steps	Anticipate certain tasks in advance
Children are admitted for health reasons; this is the route they take to get to a hospital classroom.	Enter through the health route	Admission to the hospital classroom is by health referral.
That all the services agree, that all the personnel agree and that we then move towards the same point.	Walk to the same point.	Working collaboratively to achieve a common goal

**Table 3 children-09-01395-t003:** To walks is to Learn.

Metaphorical Expression	Origin Domain Learning	Domain Goal Walk
To see where the learning difficulties of our children in treatment are and to be able to support and help them, giving them tools so that they can continue to make progress.	Children to move forward	Let the children learn
It is an articulated work so that we can help all these children with formal education.	To move forward	Children should be assisted in learning
Each student learns and progresses at their own pace.	Students progress at their own pace	Students learn at different times, according to their own abilities

**Table 4 children-09-01395-t004:** Life is War.

Metaphorical Expression	Origin Domain War	Domain Goal Life
The little school is a refuge	Refuge	The hospital school is a welcoming and safe space
A space where you are filled with the power to continue fighting	Space to keep fighting	The hospital school generates the spirit of striving for recovery.
A space where you are filled with the power to keep on winning.	Space to keep on winning	The hospital school generates encouragement to overcome adversity
From the trench that each one of us approaches in this hospital	Trench	Specialty in which everyone works to support the recovery of children’s health.
Beating cancer and getting stronger to be a professional	Beating cancer	Recovering from and surviving cancer
We as teachers and as specialists involved in their welfare have it very much in our sights.	Targeting	Permanent reminder of the family situation

**Table 5 children-09-01395-t005:** Emotions are Contagious Diseases.

Metaphorical Expression	Origin Domain Contagious Diseases	Domain Goal Emotions
It also infects the therapeutic team and the care team with that color, that light and that desire to work for children.	To spread the desire to work for children	To generate in the therapeutic and assistance team the desire to work for children.
The hospital classroom also spreads the joy of continuing to work with the children.	Conveying joy	The hospital classroom generates joy

**Table 6 children-09-01395-t006:** To Collaborate is to Hold Hands.

Metaphorical expression	Origin Domain Holding Hands	Target Domain Collaborate
Interaction is generated, so this goes hand in hand with constructive learning, that is, the student is involved in their own learning.	Goes hand in hand (with learning)	The student collaborates with his own learning.
Teachers work hand in hand with parents or caregivers with the pediatrician, head nurses.	Working hand in hand	The teachers collaborate with the medical staff
We work very closely with the Ministry of Health.	Hand in hand	The Ministries of Health and Education are working together.

**Table 7 children-09-01395-t007:** Labor is an Articulated Machine.

Metaphorical expression	Origin Domain Articulated Machine	Target Domain Job
We are working very closely with the Ministry of Health in the time of this pandemic.	Highly articulated work	Working collaboratively with the health secretariat
The Ministry of Public Health has the responsibility to provide emotional support to teachers, to provide training in biosafety and ethics standards and to work in an articulated manner.	Articulated work	They work collaboratively with the Ministry of Public Health.

**Table 8 children-09-01395-t008:** The Hospital Classroom is a Field.

Metaphorical expression	Origin Domain Field	Target Domain Hospital Classroom
The beautiful field of hospital pedagogy	Beautiful countryside	Hospital pedagogy is a rewarding job.
We have a large field to work with.	A large field	A job that generates diverse opportunities

**Table 9 children-09-01395-t009:** Affection is a Warmth.

Expresión metafórica	Dominio Origen Calor	Dominio Meta Afecto
Education, hospital pedagogy is really needed, professionals are needed to provide quality and warmth throughout the treatment.	Provide warmth	Professionals who empathize with children are needed.
On the other hand, it touches that emotional fiber that we must have, which gives warmth to health personnel.	Gives warmth	The atmosphere at the children’s hospital is comforting to the health care staff.

**Table 10 children-09-01395-t010:** To Improve is to Illuminate (Center).

Metaphorical expression	Origin Domain Illuminate	Dominio Meta Mejorar
El aula hospitalaria se volvió como el sol de la atención.	Sun of attention (illuminates)	The hospital classroom improves the schooling process.

**Table 11 children-09-01395-t011:** Life is a Play.

Metaphorical expression	Origin Domain Theater Play	Domain Goal Life
The health-disease process also has a special characteristic in which he is an actor, a protagonist.	Actor, protagonist	The health-disease process is learner-centered.
Be the protagonist of your life	Starring	To give importance to one’s own life.

**Table 12 children-09-01395-t012:** The System is a Person.

	Origin Domain Person	Target Domain System
Turn the entire system upside down so that it can respond to the educational needs of these children and youth.	Turn upside down	Change everything established in the system.

**Table 13 children-09-01395-t013:** Teachers are Tools.

Metaphorical expression	Origin Domain Tools	Target Domain Teachers
I was studying a career where I was going to be able to be an instrument for them.	To be an instrument	To be useful for the recovery of children.

**Table 14 children-09-01395-t014:** Health is an Object.

Metaphorical expression	Origin Domain Object	Meta Domain Health
Many of them experienced physical loss of family members and also obviously loss of health.	Loss of health	Become sick

## Data Availability

Not applicable.

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
