# Peer review of "Emotions in Hospital School Professionals: An Approach from Cognitive Linguistics"

_children, 2022, doi:10.3390/children9091395_

Round 1

Reviewer 1 Report

The article conforms to the formal criteria of a work of these characteristics. It presents an original and very interesting study because it focuses on understanding the emotions of professionals who work in hospital classrooms, who are subjected to high levels of emotional stress.

It uses a descriptive qualitative methodology based on the analysis of the discourse of 17 professionals. It stands out that the study participants are from 12 different countries, a fact that enriches the perspectives of the contributions and the results obtained. The line of work initiated could achieve improvement strategies on the emotional care of professionals in this field.

Author Response

We appreciate the time that the reviewers dedicated to the review of our work.
In the case of reviewer 1, there were no suggestions for changes.

Reviewer 2 Report

This is a very original paper that approaches the emotions of hospital school professionals from a cognitive linguistics perspective.

However, readers of this journal may not be familiar with cognitive linguistics, metaphors, etc., so it is necessary to explain them in more detail in the introduction.

Furthermore, I think the paper would be better if you clearly indicate the contribution of this study in the discussion section.

Author Response

We appreciate the time that the reviewers dedicated to the review of our work.
Regarding the comments provided by reviewer 2
- The first of them points out that it was necessary to explain in more detail cognitive linguistics and metaphors, in order to familiarize readers with this subject. This was done with the addition of a paragraph in the Introduction: lines 110 to 128.
- The second suggests indicating more explicitly the contribution of this study. This was incorporated in the Discussion: lines 418 to 429

Reviewer 3 Report

This research can be included in a qualitative approach to studying the meaning of the emotional expression of professionals at work in hospitals.

The idea is very interesting but it has been developed in a confusing and unclear way. The authors can justify the choice of this method and the procedure of data collection in the right manner.

The indication of the emerged metaphors is unclear and doesn't allow the replication of these results. This study does not promote its dissemination to the general public in the current form.

I suggest applying major revisions.

Author Response

We appreciate the time that the reviewers dedicated to the review of our work.
Regarding the opinion that the article has developed the idea in a "confusing" and "unclear" way, we understand that the reviewer may have a personal opinion on this matter, however, we do not share it. Regarding the justification for the choice of method, we have incorporated a paragraph (lines 154 to 160) to better explain what the reviewer suggests. He also suggests a better rationale for data collection, to which we respond in the paragraph inserted between lines 202 to 207.
Regarding the possibility of replicating the results, we should clarify that it is not feasible to replicate the results, given the nature of the study, what is possible is to replicate the study, for example, with family members, or also with professionals, in a less formal environment than a seminar, but, obviously, the results might not be the same. To believe that the results could be replicated, as the reviewer suggests, is to misunderstand the nature of the study.
The reviewer points out that this study does not promote its dissemination to the general public in its present form. Regarding this comment, we believe that the dissemination of a publication in a scientific journal to the general public will always be complex; however, in the case of this article, the language used is generic and facilitates its comprehension for any type of reader, especially considering the way it is presented. We regret that we do not agree with the reviewer's assessment, although we understand his concern.
Regarding any doubts the reviewer may have about the bibliography, we have included two texts.

Round 2

Reviewer 2 Report

I am very pleased to review this unique manuscript.

I think it is first prompt to introduce the approach from cognitive linguistics to know the emotion of the hospital school professionals.

I am looking forward to know the next study of the authors.

Thank you very much for this review this manuscript.